# VinePPO: Accurate Credit Assignment in RL for LLM Mathematical Reasoning

**Amirhossein Kazemnejad**[* 1], **Milad Aghajohari**[* 1], **Eva Portelance**[1,6],
**Alessandro Sordoni**[1,2], **Siva Reddy**[1,3,4], **Aaron Courville**[† 1,4,5], **Nicolas Le Roux**[† 1,4]

[1]Mila  [2]Microsoft Research  [3]McGill University
[4]Canada CIFAR AI Chair  [5]Université de Montréal  [6]HEC Montréal
{amirhossein.kazemnejad,aghajohm}@mila.quebec

## Abstract

Large language models (LLMs) are increasingly required to solve complex reasoning tasks, like mathematical problems, that involve multiple reasoning steps before feedback is received. Effectively identifying and prioritizing key steps by accurately assigning credit to these intermediate steps is essential for enhancing model performance. Proximal Policy Optimization (PPO), a state-of-the-art reinforcement learning algorithm for finetuning LLMs, addresses the credit assignment problem by employing value networks to predict the expected cumulative rewards of intermediate states. In this work, we identify significant limitations with this value estimation method. To address this, we propose VinePPO that leverages the flexibility of language environments to compute unbiased Monte Carlo-based estimates of the intermediate values. VinePPO consistently outperforms standard PPO, doing so more efficiently and with lower divergence from the reference model. Our findings underscore the critical importance of accurate credit assignment in LLM post-training and present a simple, yet effective solution.

## 1  Introduction

Large language models (LLMs) are increasingly employed in tasks requiring complex reasoning, such as solving mathematical problems (Trinh et al., 2024; OpenAI, 2024). In these settings, LLMs often engage in extended reasoning chains and perform numerous actions. Prioritizing steps that lead to correct solutions while downplaying erroneous ones during finetuning is essential for improving the performance and reducing unnecessary updates. This is particularly important as most reasoning steps generated by a model often do not impact its likelihood of solving the problem (Fig. 2).

This issue is known as the *credit assignment problem* in reinforcement learning (RL, Sutton and Barto 1998). Proximal Policy Optimization (PPO) (Schulman et al., 2017; Ouyang et al., 2022), the state-of-the-art algorithm for RL tuning of LLMs (Xu et al., 2024; Ivison et al., 2024; Shao et al., 2024), is a variant of actor-critic methods that utilizes a value network (*critic*) to handle credit assignment  (Bai et al., 2022, 2023; Havrilla et al., 2024). The value network is a separate model (the same size as and initialized from a pretrained checkpoint of the LLM) that learns to estimate expected cumulative future reward (value) of intermediate actions during training. PPO then uses the predicted values to measure the advantage of each action and update the model accordingly. For example, in Fig. 2, an ideal value network would assign a low value to $s_0$, where the model initially struggles, and a higher value to $s_2$ and beyond, where a critical action led to solving the problem.

Accurately predicting rewards from *a partial and incomplete response* requires the value network to grasp the space of correct solutions and predict the model's future behavior – both of which are

---

[*]Equal contribution.  [†] Equal advising.

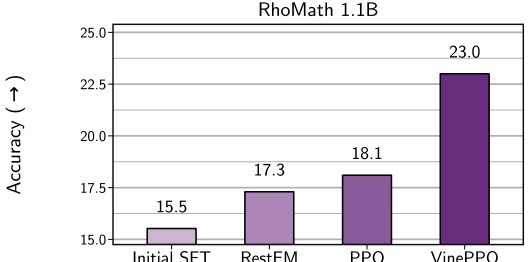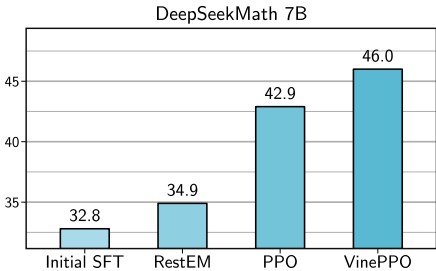

Figure 1: VinePPO outperforms standard PPO and other baselines on the MATH dataset, while also exhibiting scalability across different model sizes. The figure shows Pass@1 performance.

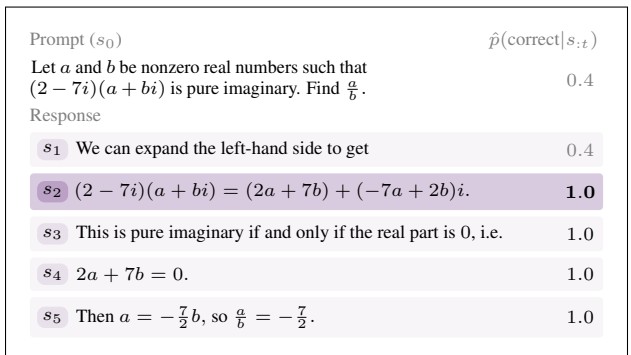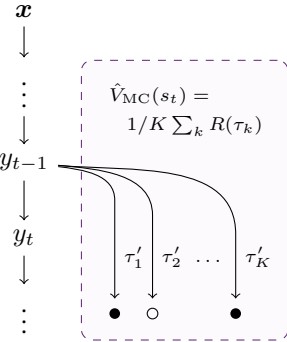

Figure 2: **(Left)** A response generated by the model. The notation $\hat{p}(\text{correct}|s_{:t})$ represents the estimated probability of successfully solving the problem at step $t$, based on nine model rollouts. In this example, only step $s_2$ is critical; after this, the model completes the solution correctly. **(Right)** Illustration of estimating the value of a state within the trajectory.

challenging. There are hints in the literature that standard PPO implementations for LLM finetuning have inaccurate value estimations. Ahmadian et al. (2024) and Luong et al. (2024) demonstrate that value networks often serve best as just a baseline in policy gradient[2]. Shao et al. (2024) shows that the value network can be replaced by averaging rewards of a group of responses to a given problem, without degradation in performance.

As estimation errors can significantly hamper model convergence and performance (Sutton et al., 1999; Greensmith et al., 2001), it is crucial to ask: *how accurately do value networks perform in practice during LLM finetuning?* While recent studies (Hwang et al., 2024; Setlur et al., 2024) have begun to highlight the importance of identifying early reasoning errors and incorporating these as training signals in "RL-free" approaches (Rafailov et al., 2023), to what extend the accuracy of credit assignment plays a role in RL tuning of LLMs remains an open question.

In this work, we evaluate the standard PPO pipeline in mathematical reasoning tasks across various model sizes and find that value networks consistently provide inaccurate estimates and a sub-optimal training signal for finetuning. To address this, we propose VinePPO. Instead of relying on value networks, VinePPO computes *unbiased* estimates by resetting the environment to intermediate states and performing independent Monte Carlo (MC) rollouts to calculate the average return of individual steps. This approach takes advantage of a special property of the language environment—the ability to easily reset to any intermediate state of a trajectory (Schulman et al., 2015). Not only does it removes the need for large, memory-intensive value networks, VinePPO also outperforms standard PPO and other baselines such as RestEM (Singh et al., 2023)(Fig. 1). VinePPO is also able to match PPO's final accuracy in fewer iterations, requiring less wall-clock time (Fig. 4), and achieving a lower KL divergence (Fig. G.3) from the base model. These findings highlight the importance of accurate credit assignment in RL post-training and position VinePPO as an effective alternative to value networks.

---

[2]setting GAE, (Schulman et al., 2016) parameter $\lambda$ to 1.

## 2 Advantage Estimation with Monte Carlo

We build on PPO (Schulman et al., 2017; Ouyang et al., 2022), for which we provide an extensive background in Appendices B and I. VinePPO only modifies the way advantages are estimated. We start by estimating the true value function $V(s_t)$. Instead of relying on a value network, for any intermediate state $s_t$, we sample $K$ independent trajectories starting from $s_t$. The average return across these trajectories serves as the value estimate:

$$\hat{V}_{\text{MC}}(s_t) := \frac{1}{K} \sum_{k=1}^{K} R(\tau_k), \quad \text{where } \tau_1, \ldots, \tau_K \sim \pi(\cdot \mid s_t). \tag{1}$$

where $\tau_k$ is an independent continuation sampled from the model, starting from $s_t$ and $R(\cdot)$ is the return over the completed trajectory. This is an MC estimate of the value function $V(s_t) = \mathbb{E}[R(\tau) \mid s_0 = s_t]$. Once the values $\hat{V}_{\text{MC}}(s_t)$ are computed, we compute the advantages with:

$$\hat{A}_{\text{MC}}(s_t, a_t) := r(s_t, a_t) + \gamma \hat{V}_{\text{MC}}(s_{t+1}) - \hat{V}_{\text{MC}}(s_t), \tag{2}$$

where $r(\cdot)$ is the step-wise reward (in practice, equal to zero unless at final step). Note that for any $K \geq 1$, the policy gradient computed using the advantage estimator $\hat{A}_{\text{MC}}$ is an unbiased estimate of the gradient of expected return.

In essence, VinePPO only alters advantage computation in PPO pipeline, leaving the rest unchanged. With this simple modification, we eliminate the need for a value network, significantly reducing memory footprint (up to 112GB for a 7B LLM) while providing unbiased estimates of advantages. The parameter $K$ offers a trade-off between computational cost (i.e. more MC samples per state) and the variance of the estimator. To enhance the efficiency of $\hat{A}_{\text{MC}}$, we also group states within a reasoning step and compute a single advantage, which is then assigned to all tokens in that step. Since everything else in the PPO pipeline of VinePPO is unchanged, by comparing the two methods, we can systematically evaluate of the impact of accurate credit assignment in RL tuning of LLMs.

## 3 Experiments

We use two strong base LLMs pretrained for mathematical reasoning: (1) DeepSeekMath 7B (Shao et al., 2024) and (2) RhoMath 1.1B (Lin et al., 2024). Our focus is the MATH dataset (Hendrycks et al., 2021), which contains competition-level problems. We compare three LLM reasoning finetuning strategies, PPO, VinePPO , and RestEM to the supervised finetuned model (SFT) baseline, from which all methods are initialized. We tune PPO hyperparameters like KL penalty coefficient, batch size, and GAE $\lambda$, applying best practices in PPO optimization. VinePPO uses the same hyperparameters as PPO but modifies the advantage estimation $A(s_t, a_t)$ to isolate the effect of accurate credit assignment. We sample $K = 9$ trajectories in $\hat{V}_{\text{MC}}$. For RestEM, we closely follow the original setup while ensuring consistency in training conditions for a fair comparison. We choose the best checkpoint based on a held-out validation set for all experiments[3].

## 4 Results and Analysis

**Task Performance** As shown in Fig. 1, VinePPO outperforms standard PPO and RestEM. The gap between VinePPO and PPO is consistent throughout the training (Fig. E.1). RestEM lacks explicit credit assignment and finetunes on full trajectories. Despite higher training accuracy, it underperforms on test, likely due to overfitting caused by training on disadvantageous intermediate steps. In addition, fig. 4 represents our ablation on $K$, observing increasing $K$ consistently improves accuracy.

**KL Divergence** The RL objective[4] aims to balance maximizing task performance while limiting deviations from the reference policy $\pi_0$, or original SFT, as measured by KL divergence. We track the KL divergence $\text{KL}[\pi_\theta \| \pi_0]$ throughout training for both methods and plot task accuracy against KL to assess this balance in Fig. G.3. The results show that VinePPO consistently achieves higher accuracy for a given KL divergence.

---

[3]Refer to Appendix D for full details.
[4]The full definition is in Appendix B.

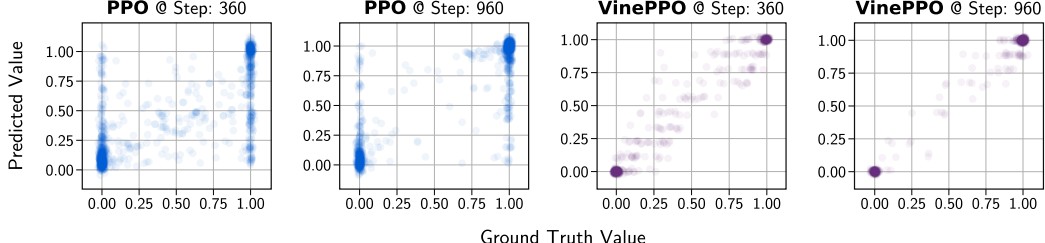

Figure 3: Distribution of predicted values for each state vs. ground truth (computed using 256 MC samples) for DeepSeekMath 7B on MATH, highlighting the nature of errors in PPO's value estimates.

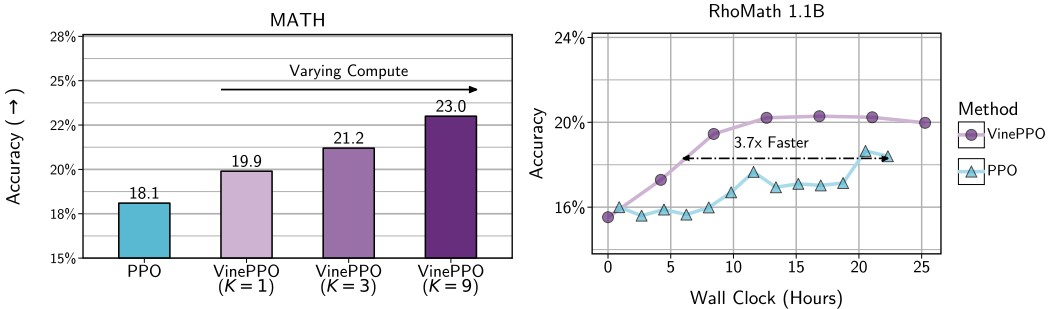

Figure 4: **(Left)** Impact of the number of sampled trajectories $K$ when estimating $\hat{V}_{\mathrm{MC}}(s_t)$, evaluated on RhoMath 1.1B models. We observe that increasing $K$ improves task performance consistently. **(Right)** Accuracy per wall clock time for both methods. Although VinePPO spend more time in each iteration, it achieves PPO's peak performance in fewer iteration and wall clock time.

**Computational efficiency** VinePPO and standard PPO need different kinds of resources. The value network needs to be trained and alongisde its optimizer consuming more GPU memory. In contrast, MC rollouts need fast inferences and as a result VinePPO is generally slower per iteration compared to PPO. In our setup, RhoMath 1.1B and DeepSeekMath 7B are 5x and 2x slower per iteration when using VinePPO . However, as shown in Fig. 4, the impact of accurate credit assignment with VinePPO is substantial. VinePPO reaches the final accuracy of PPO in fewer iterations and less time. Specifically, RhoMath 1.1B and DeepSeekMath 7B achieve PPO's final test accuracy 3.7× and 2.3× faster in wall-clock time, and in 20× and 5× fewer gradient steps, respectively.[5]

**Value Prediction Accuracy** To analyze the accuracy of value prediction, we compute the ground truth value of each state by taking 256 MC samples. We compare value network (from PPO) predictions against VinePPO's. As shown in Fig. 3, VinePPO and PPO produce errors of very different types. VinePPO estimates are unbiased, with variance peaking at 0.5 and dropping to zero at 0 and 1. In contrast, the value network's estimates exhibit high bias. See Appendix H for full details.

## 5   Related Work

**Credit Assignment in Post-Training of LLM** PPO (Schulman et al., 2017), as applied in Reinforcement Learning from Human Feedback (RLHF, Ouyang et al. 2022), was among the pioneering approaches for RL finetuning of LLMs. While effective, PPO is known for its computational overhead and sensitivity to hyperparameters. As a result, subsequent approaches have sought to simplify or bypass PPO without sacrificing performance. For example, RL-free methods such as DPO (Rafailov et al., 2023) and its newer variants (Azar et al., 2023; Ethayarajh et al., 2024) operate in a bandit setting, where the entire response is treated as a single action, without distinguishing intermediate states. Similarly, methods based on rejection sampling, like RestEM (Singh et al., 2023), finetune the model on full high-reward responses. In the realm of PPO simplifications, methods like RLOO

---

[5]Note that this is despite the fact that all of hyperparameter searches were tuned for PPO.

(Ahmadian et al., 2024) and GRPO (Shao et al., 2024) abandon the value network of PPO. They sample a group of $M$ responses per each prompt and compute the average reward (of other $M-1$ responses) as a policy gradient baseline for all tokens in the group, effectively treating the entire response as a single action. Recent works, however, have started to emphasize the importance of finer credit assignment. Work such as Hwang et al. (2024) and Setlur et al. (2024) introduce Monte Carlo-based mechanisms that detect key errors in reasoning chains and apply use them negative sample in DPO. Unlike these approaches, which rely on ad-hoc heuristics, our work fully embraces RL training pipeline and addresses the core issue of inaccurate value estimation in PPO to unlock its full potential. In parallel, there has been interest (Hosseini et al., 2024; Lightman et al., 2023a) in building better verifiers and reward models that can provide per-step feedback. Although these methods often require costly human annotation, recent efforts (Ma et al., 2023; Uesato et al., 2022; Luo et al., 2024; Wang et al., 2023) have automated data collection using MC rollouts. VinePPO is orthogonal to these approaches, as it operates within PPO-based training, optimizing a given task's reward rather than designing new reward models. Our method can further benefit from improvements in reward modeling as they emerge.

**Value Estimation in RL and Monte Carlo Tree Search** Deep RL algorithms are categorized into value-based and policy-based methods. Value-based algorithms, such as DQN and its successors (Mnih et al., 2013; Wang et al., 2015), train a neural network to predict values and derive the policy from the learned value function. Policy-based methods, including A2C, A3C (Mnih et al., 2016), SAC (Haarnoja et al., 2018), and PPO (Schulman et al., 2017), train a policy directly and use value estimates only to guide the policy updates. Typically, these methods rely on *critic networks* for value prediction. An exception is a variant of TRPO (Schulman et al., 2015), known as the *"Vine"* variant, where state value estimation is performed using MC samples. However, the authors note that the Vine variant is limited to environments that allow easy resets to any state, which is uncommon in most RL settings as the focus is on black-box engines or real-world deployment. In contrast to common RL environments, language generation, allows for easy resets to any intermediate state, presenting unique opportunities for RL tunning of LLM. In fact, when easy resets were available in RL (e.g., Go, Chess), strong MC-based methods like AlphaGo (Silver et al., 2016) and AlphaZero (Silver et al., 2017) have emerged. AlphaGo trains a policy using expert moves data and self-play, alongside a value network to predict the win probability from a given state. Then during the inference, it applies a tree search guided by MC rollouts and the value network to find the best possible moves. AlphaZero advances this approach by distilling MCTS outcomes into its policy, removing the need for expert data. Recent works have adapted AlphaZero's principles and lessons to LLM, using similar search techniques during inference to improve responses and during training to find better trajectories for distillation (Xie et al., 2024; Chen et al., 2024; Feng et al., 2023; Zhang et al., 2024; Hao et al., 2023). While this is a promising direction, VinePPO is not an MCTS method; it rather utilizes MC samples solely for value estimation and only during PPO training to improving credit assignment. In fact, inference-time search like MCTS can be layered on top of VinePPO to further enhance performance.

## 6 Conclusion

Credit assignment is a weak spot for current RL finetuning of LLMs. While value networks are tasked and trained to estimate these values, they perform poorly. VinePPO simply replaces the value networks with MC samples. We found that it reaches higher accuracy faster supporting the significant impact that accurate credit assignment has on RL finetuning of LLMs for reasoning. We hope our work encourages researchers to look into the details of RL finetuning pipelines of LLMs and to explore more computationally practical methods for accurate credit assignment.

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

# A Limitations

In this work, we focused on complex mathematical reasoning tasks, which provide a clear testbed for evaluating the impact of accurate credit assignment. While VinePPO is a general-purpose modification to PPO for LLM finetuning, its performance on more general human alignment tasks remains unclear. It is plausible that the performance gap between VinePPO and PPO would be less pronounced on tasks where the value network can generalize more easily. For example, in tasks like detecting toxicity in partial responses, the value network may perform well, reducing the advantage VinePPO offers.

# B Background

We focus on the RL tuning phase in the RLHF pipeline, following Ziegler et al. (2019a); Ouyang et al. (2022); Shao et al. (2024). In this section, we provide an overview of actor-critic finetuning as implemented in PPO.

**RL Finetuning** In this setup, the policy $\pi_\theta$ represents a language model that generates a response $\boldsymbol{y} = [y_0, \dots, y_{T-1}]$ autoregressively given an input $\boldsymbol{x} = [x_0, \dots, x_{M-1}]$, such that $\pi_\theta(\boldsymbol{y}|\boldsymbol{x}) = \prod_{t=0}^{T-1} \pi_\theta(y_t|\boldsymbol{x}; \boldsymbol{y}_{<t})$. The goal of RL finetuning is to maximize the expected undiscounted ($\gamma = 1$) finite-horizon return, while incorporating a KL-divergence constraint to regularize the policy and prevent it from deviating too far from a reference policy $\pi_0$ (typically the initial supervised finetuned (SFT) model). The objective can be written as:

$$J(\theta) = \mathbb{E}_{\boldsymbol{x}\sim\mathcal{D}, \boldsymbol{y}\sim\pi(\cdot|\boldsymbol{x})} \left[ \mathcal{R}(\boldsymbol{x}; \boldsymbol{y}) \right] - \beta \, \mathrm{KL}[\pi_\theta \| \pi_0], \tag{3}$$

where $\mathcal{D}$ is the dataset of prompts, $\mathcal{R}(\boldsymbol{x}; \boldsymbol{y})$ is the complete sequence-level reward function, and $\beta$ controls the strength of the KL penalty. Note that the policy $\pi_\theta$ is initialized from $\pi_0$.

**Language Environment as an MDP** The language generation is typically modeled as a token-level Markov Decision Process (MDP) in an actor-critic setting, where each response $\boldsymbol{y}$ is an episode. Specifically, the state at time step $t$, $s_t \in \mathcal{S}$, is the concatenation of the input prompt and the tokens generated up to that point: $s_t = \boldsymbol{x}; \boldsymbol{y}_{<t} = [x_0, \dots, x_{M-1}, y_0, \dots, y_{t-1}]$. At each time step, the action $a_t$ corresponds to generating the next token $y_t$ from fixed vocabulary. The process begins with the initial state $s_0 = \boldsymbol{x}$, and after each action, the environment transitions to the next state, $s_{t+1} = s_t; [a_t]$, by appending the action $a_t$ to the current state $s_t$. In this case, since states are always constructed by concatenating tokens, the environment dynamics are known and the transition function is *deterministic*, i.e., $P(s_{t+1}|s_t, a_t) = 1$. During the generation process, the reward $r_t$ is set to zero for all intermediate actions $a_t$'s, with the sequence-level reward $\mathcal{R}(\boldsymbol{x}; \boldsymbol{y})$ only applied at the final step when the model stops generating. A trajectory $\tau = (s_0, a_0, s_1, a_1, \dots)$ is therefore a sequence of state-action pairs, starting from the input prompt until the terminal state. Finally, we define the cumulative return of a trajectory $\tau$ as $R(\tau) = \sum_{t=0}^{T-1} r_t = \mathcal{R}(s_T) = \mathcal{R}(\boldsymbol{x}; \boldsymbol{y})$.

**Policy Gradient** Given this MDP formulation, policy gradient methods like PPO maximize Eq. 3 by repeatedly sampling trajectories and taking a step in the direction of the gradient $\boldsymbol{g}_{\mathrm{pg}} := \nabla_\theta J(\theta)$ to update the parameters. Policy gradient $\boldsymbol{g}_{\mathrm{pg}}$ takes the following form:

$$\boldsymbol{g}_{\mathrm{pg}} = \mathbb{E}_{\tau\sim\pi_\theta} \left[ \sum_{t=0}^{T-1} \nabla_\theta \log \pi_\theta(a_t|s_t) A(s_t, a_t) \right], \quad \text{where } s_t = \boldsymbol{x}; \boldsymbol{y}_{<t}, \ a_t = y_t, \tag{4}$$

$\tau = (s_0, a_0, \dots)$, and $A(s_t, a_t)$ is the *advantage* function. The gradient $\boldsymbol{g}_{\mathrm{pg}}$ points towards increasing the probability $\pi_\theta(a_t \mid s_t)$ when $A(s_t, a_t) > 0$ and the opposite when $A(s_t, a_t) < 0$. Intuitively, the advantage function $A(s_t, a_t)$ quantifies how much better taking action $a_t$ at state $s_t$ is compared to the average action taken in that state under the policy. Formally, it is defined as:

$$A(s_t, a_t) = Q(s_t, a_t) - V(s_t) = r_t + \gamma V(s_{t+1}) - V(s_t), \tag{5}$$

where $Q(s_t, a_t)$ is the state-action value and $V(s_t)$ is the per-state value function[6]. The value function, $V(s_t) : \mathcal{S} \to \mathbb{R}$, offers a long-term assessment of how desirable a particular state is under

---

[6]Such derivation is possible as the language environment is deterministic.

the current policy. Formally, it represents the expected cumulative reward obtained from starting in state $s_t$ and following the policy thereafter[7]: $V(s_t) = \mathbb{E}_{\tau \sim \pi_\theta}[R(\tau) \mid s_0 = s_t]$. PPO uses the same advantage-weighted policy gradient as in Eq. 4, but constrains policy updates through clipping to ensure stable training. For full details, see Appendix I.

**Estimating the Advantage via Value Networks**   In practice, the advantage function $A(s_t, a_t)$ is not known a priori and commonly estimated by first using a value network $\hat{V}_\phi$ to approximate the *true value function* $V(s_t)$ and then plugging the learned values into Eq. 5 or other variants such as GAE (Schulman et al., 2016). The value network is parameterized by $\phi$ and trained alongside the policy network $\pi_\theta$. The training objective for the value network minimizes the mean squared error between the predicted value and the empirical return:

$$\mathcal{L}_V(\phi) = \mathbb{E}_{\tau \sim \pi_\theta}\left[\frac{1}{T}\sum_t \frac{1}{2}(\hat{V}_\phi(s_t) - G_t)^2\right], \tag{6}$$

where $G_t = \sum_{t'=t}^{T-1} r_{t'}$ is the empirical return from state $s_t$. PPO uses the same objective for $\hat{V}_\phi$ but enhances stability by applying clipping during training (see Appendix I.1 for details). In RL-tuning of LLMs, the value network is often initialized using the initial policy $\pi_0$ (or the reward model when available), with the language modeling head swapped out for a scalar output head to predict values (Zheng et al., 2023). This setup leverages the prior knowledge of the pretrained model for value estimation.

## C   Accurate Credit Assignment with VinePPO

As outlined in Appendix B, a step in the PPO gradient update (Eq. 4) aims to increase the probability of better-than-average actions while decreasing the probability of those that perform worse—a process quantified by advantage function $A(s_t, a_t)$. However, the true advantage function is generally unknown and must be estimated, typically by substituting estimates from a value network into Eq. 5. As we will elaborate in Appendix H, neural networks are imperfect function approximators and can result in biased value estimates. Fortunately, the language environment offers a useful property that allows for deriving an unbiased estimator of value function $V(s_t)$. In this section, we first describe this property and then explain how VinePPO leverages it to enhance credit assignment.

### C.1   Language Environment

The language environment, as defined in Appendix B, possesses a unique property not commonly found in traditional RL settings: the ability to reset to any point within a trajectory. Since states are simply concatenated tokens, we can prompt the language model $\pi_\theta$ to generate continuations from any intermediate state. This flexibility allows us to explore alternative future paths from arbitrary points in a generation. In contrast, standard RL typically collect training data through sequential rollouts, a process reflected in the design of the Gym (Towers et al., 2024), the de facto RL environment API. Gym environments provide two primary functions: (1) `env.reset()`, which resets the environment to its initial state, and (2) `env.step(action)`, which advances the environment based on the agent's action. There is no mechanism for resetting to an arbitrary intermediate state within a trajectory. This design suits classic RL, where the focus is on black-box game engines or real-world deployment. Moreover, recent advancements in LLM inference engines (Kwon et al., 2023; Zheng et al., 2024) have dramatically increased the speed of on-the-fly response generation—for example, an LLM with 7B parameters can generate up to 5,000 tokens per second on a single GPU[8]. This computational efficiency makes it feasible to conduct fast environment simulation, opening up unique opportunities for RL training of LLMs.

## D   Experimental Setup

**Datasets and Pretrained LLMs**   We conduct our experiments using strong LLMs specifically pretrained for mathematical reasoning: (1) DeepSeekMath 7B (Shao et al., 2024) and (2) RhoMath

---

[7]We drop the dependency on $\pi_\theta$ for brevity.

[8]Nvidia A100 GPU with model loaded in 16bit precision.

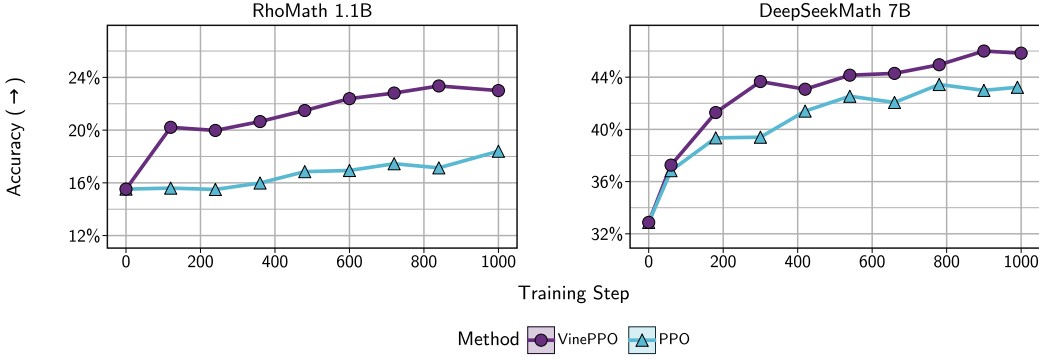

Figure E.1: Comparison of the training behavior between VinePPO and PPO. VinePPO demonstrates consistently higher accuracy (as measured on the test set of MATH dataset) throughout the training.

1.1B (Lin et al., 2024), both of which have been trained on diverse mathematical and natural language corpora. Having models from different sizes allows for evaluating the effect of scaling. We focus on mathematical reasoning datasets MATH (Hendrycks et al., 2021), which consists of competition-level mathematical problems and present a range of difficulty levels that allow for comprehensive evaluation of reasoning abilities. To ensure our setup is reproducible, we only make use of publicly available data and checkpoints on Huggingface. For each dataset, we finetune the base LLMs on their respective training sets to obtain the initial SFT models ($\pi_0$). In all experiments, we employ *full-parameter finetuning* to allow utilization of models' full capacity (Biderman et al., 2024; Sun et al., 2023).

**Evaluation** We evaluate model performance on the test sets of each dataset, using accuracy (Pass@1) as our primary metric, which measures the correctness of the final answers produced by the models. As our baseline, we adopt the standard PPO framework, as commonly implemented for LLM finetuning (Ouyang et al., 2022; Touvron et al., 2023; Huang et al., 2024). Additionally, we compare our proposed method against RestEM (Singh et al., 2023), which applies Expert Iteration, a form of Iterative Rejection Finetuning (Yuan et al., 2023; Anthony et al., 2017) with measures to prevent overfitting. All methods are initialized from the same SFT checkpoint $\pi_0$ to ensure a fair comparison.

**Hyperparameters and Training Details** To ensure standard PPO (and its value network) has a healthy training and our evaluation reflects its full potential, we first focus our hyperparameter search on PPO parameters (such as KL penalty coefficient, batch size, minibatch size, GAE $\lambda$, number of epochs per iteration) and apply all well-known techniques and best practices (Huang et al., 2024; Ivison et al., 2024) in PPO tuning (Refer to Appendix J.2 for the full list). VinePPO borrows the exact same hyperparameters from PPO and only modifies the advantage $A(s_t, a_t)$ estimation, keeping the rest of the pipeline unchanged. This allows us to isolate the effect of accurate credit assignment. We found that sampling $K = 9$ trajectories in $\hat{V}_{\mathrm{MC}}$ performs well; the effect of varying $K$ is fully analyzed in Fig. 4. For the other baseline, we closely follow the original setup while ensuring consistency in training conditions for a fair comparison. We choose the best checkpoint based on a held-out validation set for all experiments. Full implementation details, including all hyperparameters and training procedures, are provided in Appendix J.5.

## E  Training Plots

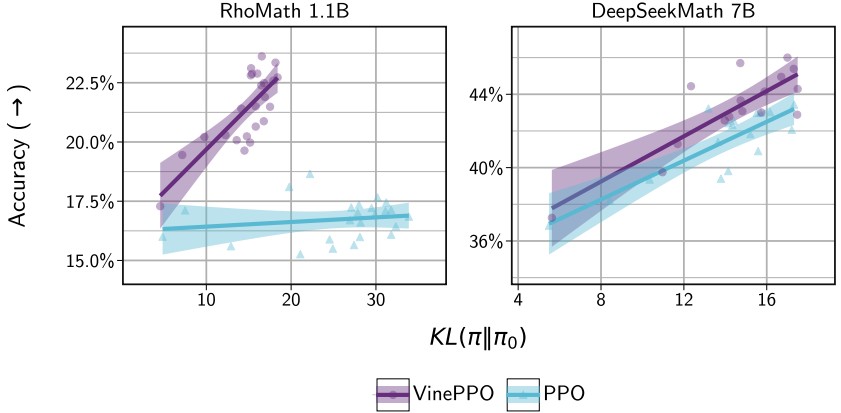

Figure G.3: Comparing task accuracy and KL divergence during training on the MATH dataset. VinePPO consistently achieves higher accuracy at similar KL levels, reflecting its more efficient credit assignment and focused updates.

## F  Temperature Tolerance

Sampling temperature is a critical hyperparameter that controls the randomness of trajectories generated by the model. At higher temperatures, the model generates more diverse trajectories, encouraging exploration that can accelerate training, especially during the early stages. However, increased diversity in the trajectories also presents a challenge: the value network in PPO must generalize over a wider range of states, complicating value estimation. To evaluate the effect of temperature on performance, we compared VinePPO and PPO runs using different temperatures $T \in \{0.6, 0.8, 1.0\}$ over 360 training steps, analysed their training dynamics. As shown in Fig. F.2, VinePPO consistently benefits from higher temperatures, achieving faster convergence and higher accuracy. In contrast, PPO not only fails to benefit from increased temperature, but also diverges when the temperature is set to its highest value, $T = 1.0$, where the trajectories are most diverse.

These findings raise concerns about the scalability of PPO, particularly in real-world scenarios involving large and diverse datasets, in contrast to VinePPO which maintains robust value estimation regardless of the diversity in the trajectories.

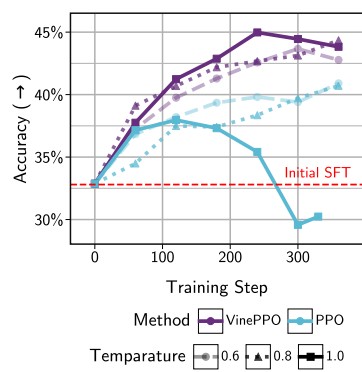

Figure F.2: Test set accuracy during training with higher temperature presented for DeepSeekMath 7B and MATH dataset. VinePPO can tolerate higher temperatures.

## G  KL Divergence

## H  Value Prediction Analysis

Both PPO and VinePPO estimate values as means to credit assignment, one employing a value network and the other using MC samples. More accurate value estimates lead to more precise advantage computations, resulting in more effective policy updates. As shown in Section 4, VinePPO consistently outperforms PPO. In this section, we explore the underlying reasons for this performance gap by closely analyzing the value prediction of both methods. To assess the accuracy of value predictions, we first establish a *"ground truth"* value for each state within trajectories, denoted as $\hat{V}^*(s_t)$, by running multiple MC rollouts (256 in our case) and averaging the returns. This provides a low-variance reference value. We then compare the value predictions in both methods against this ground truth on the DeepSeekMath 7B and MATH datasets.

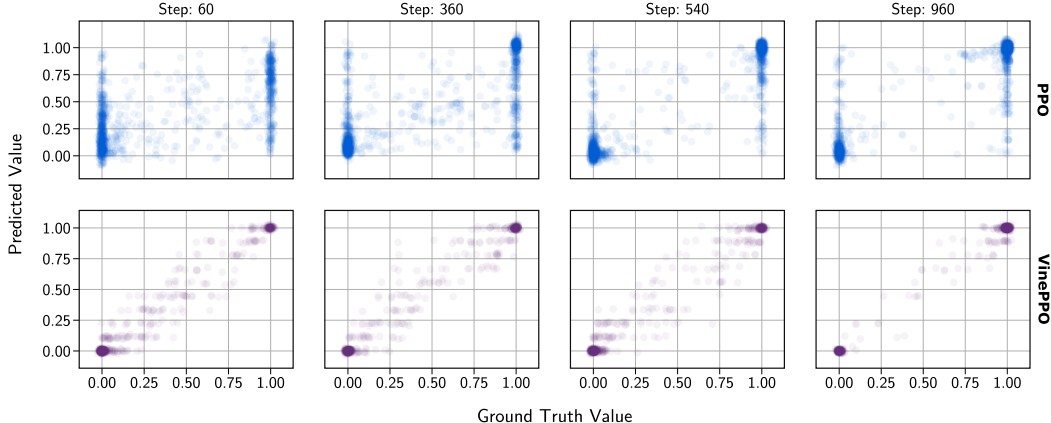

Figure H.4: Distribution of predicted values for each state vs. ground truth (computed using 256 MC samples) during training for DeepSeekMath 7B on MATH dataset, highlighting the nature of errors. VinePPO achieves much lower Mean Absolute Error (MAE).

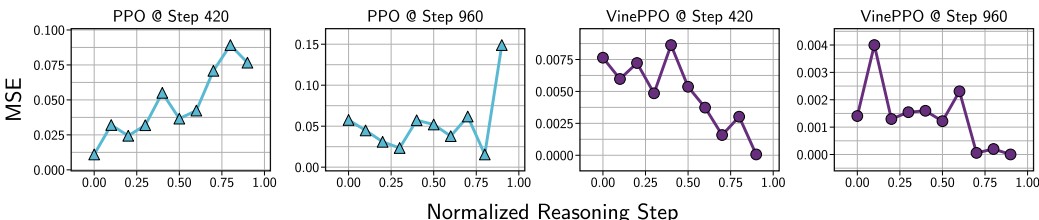

Figure H.5: Visualizing the Mean Absolute Error (MAE) of the value predictions in different point of reasoning chain. Value Network in PPO fails to generalize as the reasoning chain progresses, while VinePPO's value estimates become more accurate as the model become more deterministic.

**Accuracy** Fig. H.4 presents the distribution of value predictions during training. The errors produced by VinePPO and PPO differ significantly in their nature. VinePPO's estimates are unbiased, with variance peaking at $0.5$ and dropping to zero at $0$ and $1$. In contrast, the value network used in PPO exhibits high bias, often misclassifying bad states ($\hat{V}^*(s_t) = 0$) as good and vice versa. To further visualize accuracy, we consider a value prediction as "correct" if it falls within $0.05$ of the ground truth. The accuracy of this classification formulation is shown in Figure Fig. H.6. The value network starts with low accuracy, improving gradually to a peak of $65\%$. In contrast, VinePPO consistently achieves an accuracy of $70$-$90\%$ throughout the training process, pointing to its more reliable approach.

**Error Per Reasoning Step** To gain insights into the mechanisms behind value prediction, we analyze the prediction error at each reasoning step within a trajectory. As illustrated in Fig. H.5, PPO's value estimation error tends to increase as the reasoning chain progresses. We hypothesize this is because, at earlier steps, partial trajectories more closely resemble the training data, allowing the value network to rely on memorization. However, as reasoning progresses and the states become unfamiliar, the value network needs to generalize, where it tends to fail. In contrast, VinePPO exhibits the opposite trend: its value prediction error decreases as reasoning advances. We attribute this to the increasing determinism of later reasoning steps, which conditions on prior actions. This determinism allows the same number MC sample to provide more accurate estimates.

# I   Reviewing PPO

PPO, as used in RL tuning of LLMs, formulates language generation as token-level MDP (Appendix B), where each response $\boldsymbol{y}$ is an episode. The state at time step $t$, $s_t \in \mathcal{S}$, is the concatenation of the prompt and the tokens generated so far: $s_t = \boldsymbol{x}; \boldsymbol{y}_{<t} = [x_0, \dots, x_{M-1}, y_0, \dots, y_{t-1}]$. The

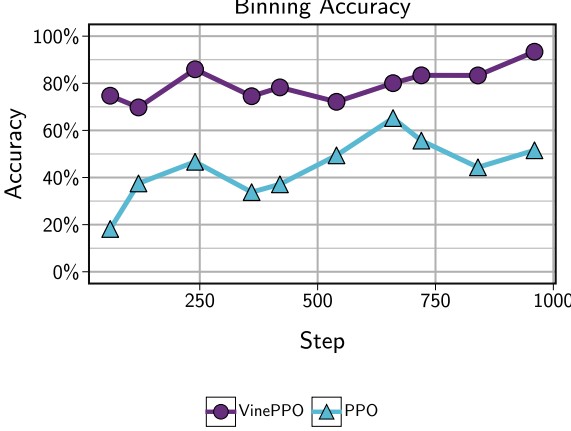

Figure H.6: Value prediction accuracy formulated as a classification problem, where a prediction is considered correct if it falls within 0.05 of the ground truth.

action $a_t$ corresponds to generating the next token $y_t$ from the model's vocabulary. Given a prompt $\boldsymbol{x}$, an episode of this MDP starts from the initial state $s_0 = \boldsymbol{x}$, and with each action taken, the environment moves to a subsequent state, $s_{t+1} = s_t; [a_t]$, by adding the action $a_t$ to the existing state $s_t$. In the language environment, because states are always formed by concatenating tokens, the environment dynamics are fully known, and the transition function is *deterministic*, meaning $P(s_{t+1}|s_t, a_t) = 1$. During the generation process, the reward $r_t$ is set to zero for all intermediate actions $a_t$'s, with the sequence-level reward $\mathcal{R}(\boldsymbol{x}; \boldsymbol{y})$ only applied at the final step when the model stops generating. Throughout the generation process, the reward $r_t$ is set to zero for all intermediate actions $a_t$, with the sequence-level reward $\mathcal{R}(\boldsymbol{x}; \boldsymbol{y})$ applied only at the final step when the model stops the generation. That is:

$$r_t = r(s_t, a_t) = \begin{cases} \mathcal{R}(\boldsymbol{x}; \boldsymbol{y}) & \text{if } t = T - 1, \text{ where } s_{t+1} = \boldsymbol{y} \text{ is terminal,} \\ 0 & \text{otherwise.} \end{cases} \quad (7)$$

A trajectory $\tau = (s_0, a_0, s_1, a_1, \dots)$ thus represents a sequence of state-action pairs that begins at the input prompt and continues until reaching the terminal state. Finally, the cumulative return of a trajectory $\tau$ is defined as $R(\tau) = \sum_{t=0}^{T-1} r_t = r_{T-1} = \mathcal{R}(\boldsymbol{x}; \boldsymbol{y})$.

The goal of RL tuning is to maximize the expected return of the model's responses to prompts in the dataset, as defined by the reward function $\mathcal{R}$ (Eq. 3). PPO, similar to other policy gradient methods, achieves this goal by repeatedly sampling trajectories for a batch of prompt sampled from $\mathcal{D}$ and taking multiple optimization steps in the direction of the gradient $\boldsymbol{g}_{\text{ppo}}$ to update the parameters. PPO gradient $\boldsymbol{g}_{\text{ppo}}$ is defined as the gradient of the following loss:

$$\mathcal{L}_{\text{ppo}}(\theta) = \mathbb{E}_{\tau \sim \pi_{\theta_k}} \left[ \sum_{t=0}^{T-1} \min \left( \frac{\pi_\theta(a_t \mid s_t)}{\pi_{\theta_k}(a_t \mid s_t)} A_t^{\theta_k}, \text{ clip}(\theta) A_t^{\theta_k} \right) - \beta \, \text{KL}[\pi_\theta \parallel \pi_0] \right] \quad (8)$$

where $\pi_{\theta_k}$ is the policy at the previous iteration, $\epsilon$ is the clipping parameter, $\beta$ is the KL penalty coefficient, $A_t^{\theta_k} = A^{\theta_k}(s_t, a_t)$ is the advantage estimate for policy $\pi_{\theta_k}$, and the $\text{clip}(\theta)$ function is:

$$\text{clip}(\theta) = \text{clip}\left( \frac{\pi_\theta(a_t \mid s_t)}{\pi_{\theta_k}(a_t \mid s_t)}, 1 - \epsilon, 1 + \epsilon \right). \quad (9)$$

Note that the KL penalty could be also added to the reward function $\mathcal{R}$. We follow the more recent implementations (Shao et al., 2024; Qwen, 2024), where it is added to the loss function. The KL term can be computed using the following unbiased estimator (Schulman, 2020):

$$\hat{\text{KL}}(\theta) = \frac{\pi_0(a_t \mid s_t)}{\pi_\theta(a_t \mid s_t)} - \log \frac{\pi_0(a_t \mid s_t)}{\pi_\theta(a_t \mid s_t)} - 1. \quad (10)$$

where $\pi_0$ denotes the reference model (initial SFT).

## I.1 Value Network

In addition to the policy $\pi_\theta$, PPO also trains a separate value network $\hat{V}_\phi$ to obtain an estimate the true values $V(s_t)$ of states $s_t$. Parameterized by $\phi$, the value network is trained alongside the policy network $\pi_\theta$ using the following loss:

$$\mathcal{L}_{\text{ValNet}}(\phi) = \frac{1}{2}\mathbb{E}_{\tau \sim \pi_\theta}\left[\frac{1}{T}\sum_{t=0}^{T-1}\max\left(\left\|\hat{V}_\phi(s_t) - G_t\right\|^2, \left\|\text{clip}(\phi) - G_t\right\|^2\right)\right] \tag{11}$$

where $\hat{V}_{\phi_k}$ is the value network at the previous iteration, $G_t = \sum_{t'=t}^{T-1}\gamma^{t'-t}r_{t'}$ is the empirical return from state $s_t$, $\epsilon'$ is a value clipping parameter, and the $\text{clip}(\theta)$ is defined as:

$$\text{clip}(\phi) = \text{clip}\left(\hat{V}_\phi(s_t), \hat{V}_{\phi_k}(s_t) - \epsilon', \hat{V}_{\phi_k}(s_t) + \epsilon'\right). \tag{12}$$

In RL-tuning of LLMs, the value network is typically initialized from the initial policy $\pi_0$ (or the reward model, if available), replacing the language modeling head with a scalar output head to predict values (Zheng et al., 2023) This approach takes advantage of the base model's prior knowledge for value estimation.

**Advantage Estimation** Once the estimated values $\hat{V}_\phi(s_t)$ are obtained, the advantages $A(s_t, a_t)$ are computed using the GAE (Schulman et al., 2016):

$$A(s_t, a_t) \approx \hat{A}^{\text{GAE}}(s_t, a_t) \tag{13}$$

$$= (1 - \lambda)\left(\hat{A}_t^{(1)} + \lambda\hat{A}_t^{(2)} + \lambda^2\hat{A}_t^{(3)} + \dots\right) \tag{14}$$

$$= \sum_{l=0}^{\infty}(\gamma\lambda)^l\delta_{t+l} \tag{15}$$

$$= \sum_{l=0}^{\infty}(\gamma\lambda)^l\left(r_{t+l} + \gamma\hat{V}_\phi(s_{t+l+1}) - \hat{V}_\phi(s_{t+l})\right) \tag{16}$$

where $\delta_t = r_t + \gamma\hat{V}_\phi(s_{t+1}) - \hat{V}_\phi(s_t)$ is the temporal difference error, $\lambda$ is the GAE parameter, and $\gamma$ is the discount factor. Also, we have:

$$\hat{A}_t^{(k)} := \sum_{l=0}^{k-1}\gamma^l\delta_{t+l} = r_t + \gamma r_{t+1} + \dots + \gamma^{k-1}r_{t+k-1} + \gamma^k\hat{V}_\phi(s_{t+k}) - \hat{V}_\phi(s_t). \tag{17}$$

Adjusting the GAE parameter $\lambda$ allows for a trade-off between bias and variance in the advantage estimates. However, as we discuss in Appendix J.5, we found that $\lambda = 1$ works best in our experiments (similar to the findings of Luong et al. (2024) and Ahmadian et al. (2024)). In this case, the GAE simplifies to the following form (assuming $\gamma = 1$): $\hat{A}^{\text{GAE}}(s_t, a_t) = \sum_{t'=t}^{T-1}r_{t'} - \hat{V}_\phi(s_t)$.

# J Experimental Details

## J.1 Datasets

We focus on mathematical reasoning datasets that require step-by-step solutions and are widely used to evaluate the reasoning capabilities of LLMs. Below is a brief overview of the datasets used in our experiments:

**MATH (Hendrycks et al., 2021)** The MATH dataset contains problems from high school math competitions, covering a wide range of topics such as algebra, geometry, and probability. For our experiments, we use the *OpenAI* split provided by Lightman et al. (2023b), which consists of 500 problems for testing and 12,500 problems for training. We further divide the training set into 11,500 problems for training and 500 problems for validation. Each problem includes a step-by-step solution, ending in a final answer marked by \boxed{} in the solution (e.g., "..*so the smallest possible value of c is $\boxed{\pi}$*"). This marking allows for verification of the correctness of model-generated responses by comparing the final answer to the ground truth. We use the scripts provided by Lewkowycz et al. (2022), Lightman et al. (2023b), and Shao et al. (2024) to extract and compare the final answers to the ground truth.

Table 1: Summary of PPO hyperparamters used in the experiments.

| Parameter | Value | |
|---|---|---|
| **TRAINING** | | |
| Optimizer | AdamW | |
| Adam Parameters $(\beta_1, \beta_2)$ | (0.9, 0.999) | |
| Learning rate | $1 \times 10^{-6}$ | |
| Weight Decay | 0.0 | |
| Max Global Gradient Norm for Clipping | 1.0 | |
| Learning Rate Scheduler | Polynomial | |
| Warm Up | 3% of training steps | |
| # Train Steps For MATH dataset | 1000 steps (around 8 dataset epochs) | |
| **GENERAL** | | |
| Maximum Response Length | 1024 tokens | |
| Maximum Sequence Length for RhoMath 1.1B | 2048 tokens | |
| Maximum Sequence Length for DeepSeekMath 7B | 2500 tokens | |
| **PPO** | | |
| # Responses per Prompt | 8 | Search Space: $\{8, 16, 32\}$ |
| # Episodes per PPO Step | 512 | Search Space: $\{256, 512\}$ |
| # Prompts per PPO Step | $512/8 = 64$ | |
| Mini-batch Size | 64 | |
| # Inner epochs per PPO Step | 2 | Search Space: $\{1, 2\}$ |
| Sampling Temperature | 0.6 | Search Space: $\{0.6, 0.8, 1.0\}$ |
| Discount Factor $\gamma$ | 1.0 | |
| GAE Parameter $\lambda$ | 1.0 | Search Space: $[0.95 - 1.0]$ |
| KL Penalty Coefficient $\beta$ | 1e-4 | Search Space: $\{$1e-1, 1e-2, 3e-3, 1e-4$\}$ |
| Policy Clipping Parameter $\epsilon$ | 0.2 | |
| Value Clipping Parameter $\epsilon'$ | 0.2 | |

Table 2: Summary of RestEM hyperparamters used in the experiments.

| Parameter | Value | |
|---|---|---|
| **TRAINING** | | |
| Optimizer | AdamW | |
| Adam Parameters $(\beta_1, \beta_2)$ | (0.9, 0.999) | |
| Learning rate | $1 \times 10^{-6}$ | |
| Weight Decay | 0.0 | |
| Max Global Gradient Norm for Clipping | 1.0 | |
| Learning Rate Scheduler | Polynomial | |
| Warm Up | 3% of training steps | |
| **RESTEM** | | |
| # iterations | 10 | |
| # Sampled Responses per Prompt | 8 | Search Space: $\{8, 32\}$ |
| Sampling Temperature | 0.6 | Search Space: $\{0.6, 0.8, 1.0\}$ |
| Checkpoints every # iteration | 500 step | |
| Checkpoint Selection | until validation improves | |
| | Search Space: {until validation improves, best validation} | |

## J.2 PPO Implementation

To ensure our PPO implementation is robust, and our evaluation reflects its full potential, we have applied a set of well-established techniques and best practices from the literature (Huang et al., 2024; Ivison et al., 2024; Zheng et al., 2023). Below, we outline the key implementation details that were most effective in our experiments:

- **Advantage Normalization**: After calculating the advantages, we normalize them to have zero mean and unit variance, not only across the batch but also across data parallel ranks. This normalization step is applied consistently in both our PPO and VinePPO implementations.

- **Reward Normalization**: We follow Ivison et al. (2024) and do not normalize the rewards, as the reward structure in our task is already well-defined within the range of $[0, 1]$. Specifically, correct responses are assigned a reward of 1, while incorrect responses receive 0.

- **End-of-Sequence (EOS) Trick**: As detailed in Appendix I, rewards are only applied at the final token of a response, which corresponds to the EOS token when the response is complete. For responses that exceed the maximum length, we truncate the response to the maximum length and apply the reward to the last token of the truncated sequence. We also experimented with penalizing truncated responses by assigning a negative reward (-1), but this did not lead to performance improvements.

- **Dropout Disabling**: During the RL tuning phase, we disable dropout across all models. This ensures that the log probabilities remain consistent between different forward passes, thereby avoiding stochastic effects that could hurt training stability.

- **Fixed KL Coefficient** We use a constant coefficient for the KL penalty. Although the original PPO implementation for finetining language models (Ziegler et al., 2019b) utilized an adaptive KL controller, more recent implementations typically do not use this approach (Ouyang et al., 2022; Touvron et al., 2023; Xu et al., 2024).

## J.3 SFT Models

To ensure a systematic and reproducible evaluation, we create our SFT models $\pi_{\mathrm{ref}}$ by finetuning the *base pretrained LLMs* (as opposed to their "Instruct" version) on the training splits of the respective datasets. Specifically, we produce two distinct SFT models: two base LLM (DeepSeekMath 7B and RhoMath 1.1B ) across MATH. The base models are finetuned using the Adam optimizer without weight decay. We employ a learning rate warm-up over 6% of the total training steps. Each model is trained for three epochs with a batch size of 64, and the best checkpoint is selected based on validation accuracy. For each SFT model, we conduct a hyperparameter sweep over learning rates in the range $\{1 \times 10^{-7}, 3 \times 10^{-7}, 1 \times 10^{-6}, 3 \times 10^{-6}, 1 \times 10^{-5}, 3 \times 10^{-5}, 8 \times 10^{-5}, 1 \times 10^{-4}\}$ to ensure optimal performance. We then use these SFT models as the initial checkpoint for training the methods mentioned in our paper.

## J.4 Evaluation

We evaluate each method's performance on the test sets of each dataset. For example, when we report that PPO achieves 42.8% accuracy on the MATH dataset for the DeepSeekMath 7B model, this means the PPO training was initialized with the SFT model specific to DeepSeekMath 7B on the MATH dataset, and accuracy was measured on the MATH test set. Our primary evaluation metric is accuracy, specifically $\mathrm{Pass@1}$, which reflects the percentage of correct responses on the first attempt. This metric is crucial because it represents a realistic user interaction, where the model is expected to deliver a correct answer without the need for multiple tries. For each evaluation, we sample a response from the model for a given prompt, using a maximum token length of 1024 and a temperature of 0.35. A response is considered correct if its final answer matches the ground truth final answer, as detailed in Appendix J.1. Furthermore, each accuracy score is averaged over 16 evaluation rounds, each conducted with different random seeds. This will ensure a robust and low variance assessment of model performance.

## J.5 Hyperparameters

In this section, we present a comprehensive overview of the hyperparameters used in our experiments.

**PPO** Finetuning LLMs using PPO is known to be sensitive to hyperparameter selection, and finding the optimal settings is critical for achieving strong performance. To ensure the robustness of our study, we explored hyperparameter values reported in recent studies (Shao et al., 2024; Zheng et al., 2023; Ivison et al., 2024; Huang et al., 2024) and conducted various sweeps across a wide range of values to identify the best configuration for our tasks and models. The full set of hyperparameters, along with their respective search spaces, is detailed in Table 1.

Table 3: Average time spent per each training step for different methods and models measured for MATH dataset

| Method | Model | Hardware | Average Training Step Time (s) |
|---|---|---|---|
| PPO | RhoMath 1.1B | $4 \times$ Nvidia A100 80GB | 80 |
| VinePPO | RhoMath 1.1B | $4 \times$ Nvidia A100 80GB | 380 |
| PPO | DeepSeekMath 7B | $8 \times$ Nvidia H100 80GB | 312 |
| VinePPO | DeepSeekMath 7B | $8 \times$ Nvidia H100 80GB | 583 |

**VinePPO** We utilized the same hyperparameter setup as in the PPO implementation (Table 1) for VinePPO. The number of MC samples, $K$, was set to 9 for all experiments.

**RestEM** To ensure fair comparison we equalize the number of sampled responses for training between our RestEM run and our PPO runs. Therefore, in each RestEM iteration we sample 8 responses per prompt and train for 8 epochs on the correct responses. In order to boost RestEM's performance we also run a sweep on other sensible parameters but we noticed no improvement (Table 2).

### J.6 Compute

All experiments were conducted using multi-GPU training to efficiently handle the computational demands of large-scale models. For the RhoMath 1.1B model, we utilized a node with $4 \times$ Nvidia A100 80GB GPUs to train both PPO and VinePPO. For the larger DeepSeekMath 7B model, we employed a more powerful setup, using a node with $8 \times$ Nvidia H100 80GB GPUs. Additionally, for training DeepSeekMath 7B models with the RestEM approach, we used a node with $4 \times$ Nvidia A100 80GB GPUs. The average training step time for each method on the MATH dataset is presented in Table 3.

### J.7 Software Stack

Both PPO and VinePPO require a robust and efficient implementation. For model implementation, we utilize the Huggingface library. Training is carried out using the DeepSpeed distributed training library, which offers efficient multi-GPU support. Specifically, we employ DeepSpeed ZeRO stage 0 (vanilla data parallelism) for RhoMath 1.1B and ZeRO stage 2 (shared optimizer states and gradients across GPUs) for DeepSeekMath 7B . For trajectory sampling during RL training, we rely on the vLLM library (Kwon et al., 2023), which provides optimized inference for LLMs. Additionally, VinePPO leverages vLLM to generate Monte Carlo samples for value estimation. This software stack ensures that our experiments are both efficient and reproducible. For instance, during VinePPO training, we achieve an inference speed of up to 30K tokens per second using $8 \times$ Nvidia H100 GPUs with the DeepSeekMath 7B model.

### J.8 Reproducibility

In this study, all experiments were conducted using open-source libraries, publicly available datasets, and open-weight LLMs. To ensure full reproducibility, we will release both Singularity and Docker containers, pre-configured with all dependencies and libraries, enabling our experiments to be run on any machine equipped with NVIDIA GPUs, now or in the future. Additionally, we will make our codebase publicly available on GitHub at `https://www.omitted.link`.

