# OpenReview forum: "VinePPO: Accurate Credit Assignment in RL for LLM Mathematical Reasoning"
_NeurIPS.cc/2024/Workshop/MATH-AI — MATH-AI 24_

### Official Review · Reviewer_nosb · 2024-10-06

**Rating:** 7
**Confidence:** 4

**Review:**

The paper proposes VinePPO, which replaces the value network of PPO with the Monte-Carlo estimation of the value function.

Pros:
 - The paper is well-motivated: during PPO training of the language models, there are signs that the value network cannot provide accurate estimate of the value functions. The authors also carry out experiments to justify the claim.
 - The authors show positive improvements of the proposed VinePPO algorithm to the original PPO algorithm on math reasoning datasets with two models (RhoMath 1.1B & DeepSeekMath 7B).

Cons:
 - The proposed method seems to be a universal method that can also extend to general alignment scenarios. The paper will benefit from more experiments on general alignment tasks (e.g. consider testing on AlpacaEval). However, it will be hard to naturally partition the language model's generation into states of multiple tokens. This seems to limit the proposed algorithm to reasoning tasks and models with output format of explicit steps. (The authors already mentioned this limitation in Appendix B.)
 - The paper will benefit from comparing with other baseline policy optimization algorithms without a value network, e.g., the GRPO algorithm (Shao et al. 2024) that the authors claimed to have motivated their work.

---

### Official Review · Reviewer_FGTg · 2024-10-06
**This paper considers the credit assignment problem of RL fine tuning in LLMs. It proposes a new algorithm based on proximal policy gradient to address the limitations of current method and provides empirical numerical experiments to show its performance.**

**Rating:** 6
**Confidence:** 3

**Review:**

Summary:
This paper addresses one crucial problem in LLM with reinforcement learning fine tuning, the credit assignment problem. Based on Proximal Policy Optimization (PPO), the author proposed a new method VinePPO based on Monte Carlo rollouts instead of value networks. By numerical experiments, the author showed that VinePPO overwhelms standard PPO in math task datasets with respect to accuracy and concentration.

Strengths:

1. The empirical analysis of the algorithm are in-depth with detailed illustration. The paper is well-written and to some degree self-contained.

2. The difference between PPO and VinePPO is only using MC instead of value networks, yet the adjusted algorithm could deal with credit assignment issue.

Weaknesses:

1. Lack of theoretical insights of intuitive explanation (see questions below) about the algorithm or about existing principles regarding what procedure would be suitable for credit assignment. I believe with such math intuition, the paper will be more convincing.

2. I suggest the author providing more comparison between the proposed algorithm and other fine tuning strategies, which could make the paper more convincing.

Questions: I’m not an expert in LLM empirical study and fine-tuning so please forgive me if my questions are naive.

1. May one ask if you have any theoretical intuition about why value network provides inaccurate estimation in credit assignment?

2. Are there any math intuition about why MC algorithm performs much better apart from empirical illustration? In other words, what's your intuition about choosing this fine-tuning strategy?

3. Apart from math tasks, could your algorithm be applied to other kinds of tasks which need fine tuning? Thanks!

---

### Official Review · Reviewer_863D · 2024-10-07

**Rating:** 6
**Confidence:** 4

**Review:**

**Overall**:

This paper introduces a novel approach for credit assignment in large language models (LLMs) using a modified version of PPO, named VinePPO. The authors highlight an issue with potentially high bias when using classical PPO for credit assignment in LLM reasoning and propose leveraging Monte Carlo-based value estimates instead of a value network for more accurate value estimation, leading to improved performance.

The paper is generally well-written, with clearly articulated points and promising simulation results. However, providing a deeper explanation of the underlying mechanisms driving the improved performance of VinePPO would enhance the reader's understanding. Specifically, elaborating on why the classical PPO lead to high bias, and the theoretical or empirical justifications for why it leads to better credit assignment by using VinePPO, would offer clearer insights into the advantages of this approach.

**Strength**:

The paper presents a clear motivation and offers a neat approach to addressing inaccuracies in credit assignment for LLM reasoning. The writing is clear and easy to follow, with a well-structured presentation. The extensive simulation studies are thorough, and the hyper-parameter tuning process is well-explained, contributing to a solid overall contribution.

**Weakness**:
1. While the results are promising, the use of Monte Carlo for unbiased estimation is a well-established concept in the context of reinforcement learning (RL) and has been widely applied. This might moderate the originality of the paper.

2. The explanation of why VinePPO outperforms classical PPO remains somewhat unclear. Expanding on the rationale behind its superiority, as touched upon briefly in Lines 512-520, would improve the reader's understanding. A more detailed elaboration on the theoretical and empirical aspects of this improvement would be beneficial in the main paper.


**Questions**:

1. The results presented in the paper are quite promising. However, does VinePPO consistently outperform PPO only in the specific context of LLM reasoning, or does this advantage extend to other tasks, such as general language modeling or even non-language environments? It would be interesting to hear the authors' thoughts and speculations regarding its generalizability.

3. In terms of the tradeoff between accuracy and computational complexity introduced by VinePPO, would it be possible to combine PPO and VinePPO depending on the training stages? Monte Carlo sampling might offer more stable value estimates early in training, where VinePPO could be most effective. Later, when the training process stabilizes, switching to classical PPO could reduce computational overhead. I wonder how the authors feel about this potential hybrid approach, and if there’s any comparison between the accuracy of PPO and VinePPO with respect to the number of training iterations.

**Minor issue**:
1. While this may be a matter of terminology, it could be somewhat misleading to describe classical PPO as "biased" in Line 420. As the algorithm's recursion size increases, PPO provides consistent estimates of the value function under proper conditions. It may be more appropriate to state that "VinePPO" yields more accurate value function estimates compared to classical PPO.

---

### Official Review · Reviewer_D8BP · 2024-10-08
**Smart observation**

**Rating:** 7
**Confidence:** 3

**Review:**

This paper presents a highly insightful observation. While Proximal Policy Optimization (PPO) often struggles with accurately estimating intermediate state values, the proposed method cleverly capitalizes on a unique feature of the language environment—the ability to easily reset to any intermediate state in a trajectory. By leveraging this property, the authors utilize Monte Carlo rollouts to estimate these values more effectively.

Pros:

The use of Monte Carlo rollouts to estimate intermediate state values is both innovative and well-suited to the characteristics of the language environment. This novel approach addresses a key limitation of PPO, demonstrating creative problem-solving.

The proposed method demonstrates a significant improvement over the baseline, showcasing its effectiveness in addressing the problem at hand.

Cons:

The explanation of Figure 3 is unclear, making it difficult to interpret its significance. I recommend providing additional context or clarification to help readers better understand its role in supporting the paper’s claims.

The paper lacks comparisons with other existing methods, which makes it difficult to evaluate how the proposed approach performs relative to the broader state of the art.

---

### Decision · Program_Chairs · 2024-10-09

**Decision:**

Accept

**Comment:**

The paper identifies a weakness of PPO and proposes an effective alternative that can be generally applied.